# Extracting Symbolic Sequences from Visual Representations via Self-Supervised Learning

## Abstract

In this paper, we explore the potential of abstracting complex visual information into discrete, structured symbolic sequences using self-supervised learning (SSL). Inspired by how language abstracts and organizes information to enable better reasoning and generalization, we propose a novel approach for generating symbolic representations from visual data. To learn these sequences, we extend the DINO framework to handle both visual and symbolic information. Initial experiments suggest that the generated symbolic sequences capture a meaningful level of abstraction, though further refinement is required. An advantage of our method is its interpretability: the sequences are produced by a decoder transformer using cross-attention, allowing attention maps to be linked to specific symbols and offering insight into how these representations correspond to image regions. This approach lays the foundation for creating interpretable symbolic representations with potential applications in high-level scene understanding.

## 1 Introduction

In recent years (Bengio et al., 2013; Krizhevsky et al., 2017), advances in computer vision and machine learning have significantly improved our ability to learn from complex visual data. However, most learned representations remain continuous and unstructured(LeCun et al., 2015), making it difficult to reason about high-level abstractions and relationships in the data. Inspired by language structure, which allows us to abstract and generalize from perceptual input, we explore whether it is possible to generate discrete, structured symbolic representations from visual data through self-supervised learning (SSL).

Language provides a compelling framework for abstraction(Bisk et al., 2020): it captures meaning through compositional symbols that represent and generalize complex information, enabling higher-level reasoning. Translating this capacity to machine learning could be a key step toward more interpretable and generalizable models(Lake et al., 2016). However, current approaches to visual representation learning primarily focus on learning dense, continuous features, which lack the compositional properties needed for symbolic reasoning. This gap motivates our investigation into generating symbolic sequences from visual input, where structured symbols can encapsulate the variations and complexities of visual data in a compact, interpretable form.

In this work, we introduce a novel approach to generating symbolic representations from visual data using an extended version of the DINO framework(Caron et al., 2021; Oquab et al., 2024), which is designed to handle both visual and symbolic information. Our method leverages pre-trained visual representations from a Vision Transformer (ViT)(Dosovitskiy et al., 2021) and extends them to produce symbolic sequences that can abstract the compositional properties of visual scenes. By utilizing a decoder transformer with cross-attention mechanisms, we ensure that the generated symbols can be interpreted and linked to specific regions in the input data, providing a more interpretable model.

Our main contributions are as follows: (1). We propose a novel method for generating structured symbolic sequences from visual data through self-supervised learning, inspired by linguistic abstraction. (2). We extend the DINO framework to handle both visual and symbolic information, enabling the learning of compositional symbolic representations. (3). We demonstrate the interpretability of our method by linking the generated symbols to visual regions through attention maps, offering insights into how these symbols correspond to image features. (4). Initial experiments show that

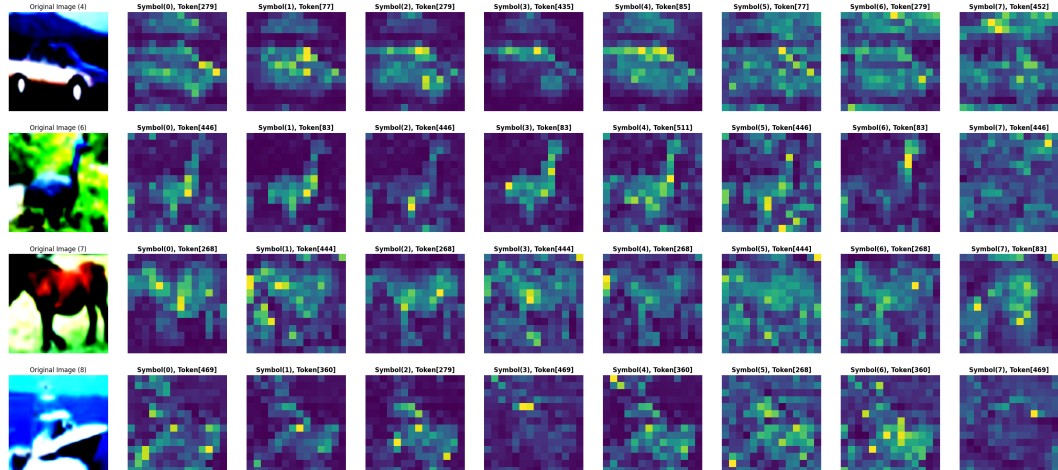

Figure 1: Visualization of four sample images alongside their generated sequences and corresponding attention masks, produced by our model. The sequences are generated using a temperature-softmax discretization process with a temperature of 0.12 during training. Attention masks, associated with each sequence element, are extracted from the cross-attention layers in the deepest layer of the descriptor module. From left to right: The first column shows the input sample images, followed by the generated sequences and their corresponding attention masks.

our approach captures a meaningful level of abstraction, though further refinement is needed. This suggests potential for high-level scene understanding and interpretability.

By bridging the gap between continuous visual representations and discrete symbolic reasoning, our approach opens the door to more interpretable models. It lays the groundwork for further exploration in abstract visual understanding.

## 2 RELATED WORK

**Visual Representation Learning and Discrete Latent Representations** Self-supervised learning (SSL)(Chen et al., 2020; Radford et al., 2021) has led to significant progress in visual representation learning, with Vision Transformers (ViT) proving particularly effective at extracting meaningful features by attending to different regions of an image. Methods like DINO build on ViT to capture dense, continuous representations from visual data. However, while these representations are powerful, they often lack the structure necessary for symbolic reasoning and high-level abstractions. A growing body of work addresses this limitation by introducing discrete latent representations(van den Oord et al., 2018; Yu et al., 2022), which transform continuous visual features into discrete codes or tokens. These discrete representations offer a more structured and interpretable way to model complex visual inputs. However, while they provide compact encodings, they typically do not impose specific structural properties or constraints on the learned codes, leaving the challenge of generating meaningful, compositional symbolic sequences open. Our work builds on this by adopting discrete representations and focusing on learning structured symbolic sequences that capture the underlying compositional nature of visual data.

**Image Captioning and Encoder-Decoder Models** Traditional image captioning models typically follow an encoder-decoder architecture(Vinyals et al., 2015; Xu et al., 2016), where the encoder (often a CNN or Vision Transformer) processes the image into a dense, continuous representation, and the decoder (such as an RNN or Transformer)(He et al., 2015; Vaswani et al., 2023) generates descriptive language sequences based on these features. These models rely on explicit supervision, using labelled datasets with human-annotated captions to guide the mapping from images to text. However, our approach differs fundamentally. While we also employ a standard encoder-decoder architecture with a pretrained Transformer to capture visual features, our method does not rely on human-provided labels, predefined language targets, or prior training on linguistic models.

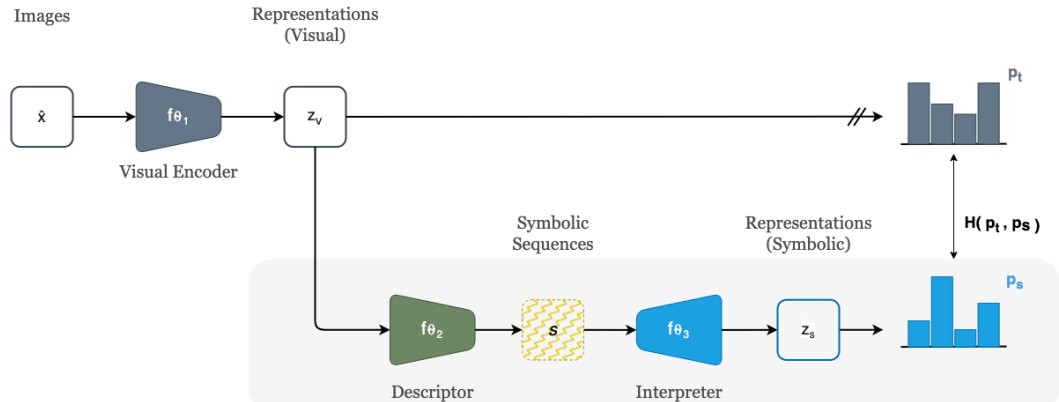

Figure 2: Schematic drawing of the teacher-student setup. The teacher model consists as usual of an encoder and projector, while the student models consist of a decoder and encoder plus the regular projector. The input images are passed to a pretrained teacher, and the representations of it are then fed to the student. finally, the outputs of the projectors are compared. The student weights are then adjusted to mimic the output of the teacher. In our experiments, we work under the assumption of an existing visual encoder and focus solely on training the projector layer of the teacher using EMA while keeping the rest of it frozen.

Instead, we autonomously generate symbolic sequences directly from visual data in a self-supervised manner, discovering structured outputs without the guidance of any external symbolic or linguistic framework. This makes our task more challenging and allows for the emergence of abstract and compositional visual representations in a fully unsupervised setting.

**Neuro-Symbolic Learning and Symbolic Reasoning** Neuro-symbolic approaches(Susskind et al., 2021) aim to combine the pattern recognition capabilities of neural networks with the logical, interpretable reasoning strengths of symbolic AI. These models use neural networks to process raw data (such as images or text) and output symbolic representations or logical rules for high-level reasoning tasks. Recent advances have introduced methods for generating symbolic sequences directly from visual data, allowing models to infer perceptual features and symbolic abstractions in a unified framework. Our work aligns with this direction by generating symbolic sequences from visual input, directly integrating perception with structured symbolic reasoning without requiring external symbolic systems.

**Interpretability through Attention Mechanisms and Discrete Representations** Attention mechanisms, particularly in Transformer architectures, have proven essential for improving the interpretability of models by highlighting which parts of the input data are most relevant for a given prediction. In visual models, attention maps make it possible to understand how specific regions of an image contribute to the output. Some approaches (Zhang et al., 2021) go further by tokenizing images into discrete units, which can then be mapped to symbolic representations, offering a more interpretable and structured view of the visual data. Our method builds on this by using cross-attention to link symbolic sequences to specific image regions, ensuring that the learned representations are not only abstract and symbolic but also interpretable, providing transparency in how the model processes and understands visual scenes.

## 3 METHOD

Our proposed method follows a teacher-student framework(Hinton et al., 2015) for generating symbolic representations from visual data, incorporating cross-attention mechanisms, discretization strategies, and symbolic token embedding. The teacher network, which uses a pretrained Vision Transformer (ViT), provides stable visual representations, while the student network is trained to generate symbolic sequences that approximate these representations. We outline the method in four

major components: (1) Teacher-Student Framework, (2) Symbolic Token Discretization and Embedding, (3) Training Procedure, and (4) Exploration Strategies.

## 3.1 TEACHER-STUDENT FRAMEWORK

The core of our approach is a **teacher-student framework**, where the teacher provides pretrained visual features, and the student learns to represent these features symbolically. The student generates symbolic sequences using a decoder and re-embeds them into a joint distribution space through an encoder. We will now describe each network in detail.

**Teacher Network** The teacher network $T$ consists of an encoder module $E_T$, which is a pretrained Vision Transformer (ViT-B/16) from the DINO method, and a projector module $P_T$. Given an input image $x$, the teacher's encoder produces a visual representation $z_t$:

$$z_t = E_T(x),$$

which is then projected by the teacher's projector $P_T$ into a joint distribution space:

$$\mathbf{p}_t = P_T(z_t).$$

The teacher network is frozen during training, except for the projector head $P_T$, which is updated using an **Exponential Moving Average (EMA)** of the student projector weights:

$$\theta_{P_T} \leftarrow \lambda\theta_{P_T} + (1 - \lambda)\theta_{P_S},$$

where $\lambda$ is the EMA decay factor, $\theta_{P_T}$ denotes the teacher's projector weights, and $\theta_{P_S}$ denotes the student's projector weights.

**Student Network** The student network $S$ is composed of three main components: a decoder module $D_S$, an encoder module $E_S$, and a projector module $P_S$. Unlike the teacher, the student is initialized randomly and trained to align its representations with the teacher's by generating symbolic sequences that abstract visual information. The student model operates in two phases: symbolic sequence generation and embedding alignment.

- **Symbolic Sequence Generation**: To generate descriptions of varying levels of detail, the descriptor $D_S$ autoregressively transforms the teacher's visual representations $z_t$ into symbolic sequences $s_s$ using cross-attention mechanisms. These sequences represent high-level semantic abstractions, capturing key features of the input. By generating symbolic sequences of different lengths for the same scene, shorter sequences focus on broad semantic features, while longer sequences capture more specific details. This approach encourages the student model to learn a general-to-specific behavior, enabling it to adjust to different levels of abstraction in the data.

$$\mathbf{s_s} = D_S(\mathbf{z}_t),$$

- **Discretization and Re-Embedding**: To ensure gradients can propagate through the symbolic sequence $\mathbf{s}_s$, a discretization process is applied over the token embeddings rather than performing a hard, non-differentiable operation. For each logit in the sequence, an approximation to the maximum is computed to assign a token, which is then mapped to its corresponding embedding. This approach allows the model to maintain trainability while preserving the symbolic nature of the sequence. By discretizing $\mathbf{s}_s$ into distinct tokens, we ensure that each token represents a unique and well-defined semantic concept, avoiding blended or ambiguous representations. These discrete tokens are then re-embedded into continuous representations $z_s$ through the interpreter, creating meaningful embeddings aligned with the symbolic abstractions. Finally, these embeddings are processed by the student's projector, $P_S$, to map them into a joint distribution space, facilitating alignment with the teacher's representations:

$$\mathbf{p}_s = P_S(E_S(\mathbf{s}_s)),$$

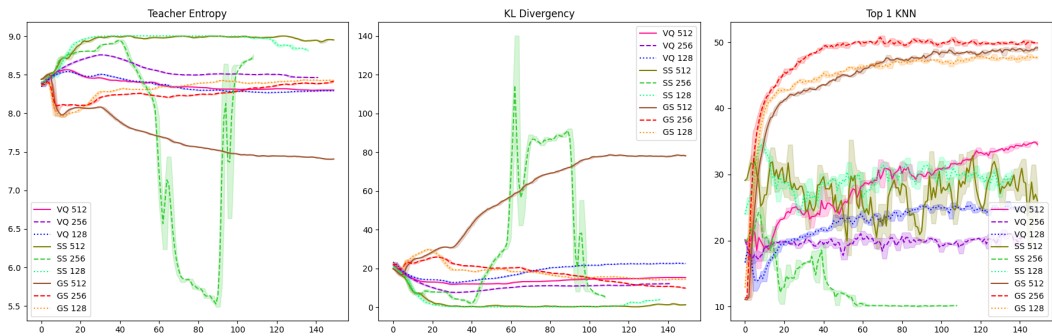

Figure 3: Training process of nine variations of our method, including three discretization variations with varied vocabulary sizes in symbolic descriptions. From left to right: (a) shows the teacher entropy over training steps; (b) displays the KL divergence between teacher and student distributions; (c) presents the evaluation performance using a k-NN metric across the different variations.

## 3.2 SYMBOLIC TOKEN DISCRETIZATION AND EMBEDDING

Discretization is crucial to our method, as it converts continuous visual representations into structured symbolic sequences. We explore three different discretization strategies to evaluate the effectiveness of the symbolic abstraction:

1. **Low-Temperature Softmax**: To approximate a maximum operation, we apply a softmax function with a very low temperature, which selects the most probable token for each step in the sequence.

$$\mathbf{s}_q = \text{softmax}\left(\frac{\mathbf{s}_s}{\tau}\right), \quad \text{with} \quad \tau \to 0.$$

2. **Gumbel Softmax**: The Gumbel-Softmax (Jang et al., 2017) trick samples discrete tokens while maintaining differentiability, allowing backpropagation through discretization.

$$\mathbf{s}_q = \text{GumbelSoftmax}(\mathbf{s}_s, \tau).$$

3. **Vector Quantization (VQ)**: In this variant, we apply a Vector Quantization (VQ) layer over the continuous output of the decoder, which maps each continuous output to the nearest code in a fixed codebook.

$$\mathbf{s}_q = \arg\min_{e_i \in \mathcal{C}} \|\mathbf{s}_s - e_i\|_2,$$

where $\mathcal{C}$ is the codebook of quantized vectors.

Once the sequence is discretized, we embed the symbolic tokens $\mathbf{s}_q$ through an encoder-only transformer $E_S$. To encourage compositionality, we split the sequence into subsequences of increasing length (powers of two). We start with a sequence of length 1 and progressively generate subsequences of lengths 2, 4, and 8, where each subsequence $\mathbf{s}_q^{[:n]}$ contains all previous elements. For each subsequence, we obtain an embedding:

$$\mathbf{p}_s^{(n)} = P_S(E_S(\mathbf{s}_q^{[:n]})) \quad \text{for } n = 1, 2, 4, 8.$$

The final joint distribution is the aggregated sum of the subsequences:

$$\mathbf{p}_s = \sum_{n=1}^{8} \mathbf{p}_s^{(n)}.$$

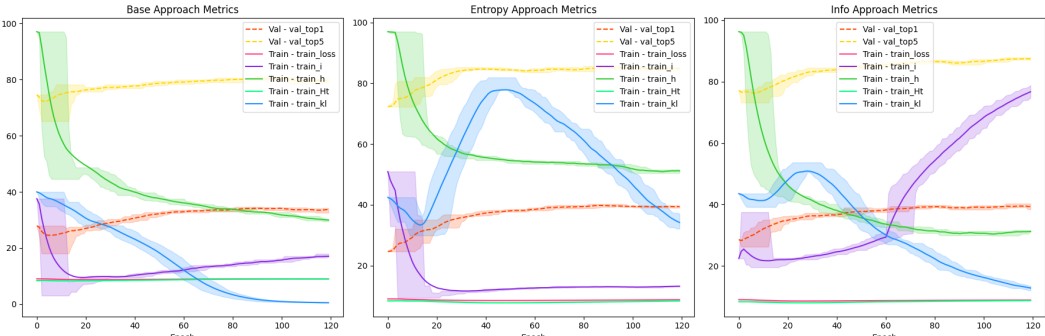

Figure 4: Training curves for three exploration strategies: (a) Base Strategy, (b) Entropy Encouragement Strategy, and (c) Information Maximization Strategy. Each plot tracks multiple metrics over training steps: top-1 and top-5 classification accuracy (probing), training loss, teacher entropy, KL divergence between teacher and student distributions, information content of the generated sequences, and entropy of the logits from the decoder transformer. These metrics reflect the effects of the different strategies on exploration, variability, and performance during the symbolic sequence generation process.

## 3.3 TRAINING PROCEDURE

We follow a self-supervised learning approach, where the teacher visual encoder generates visual representations using images from CIFAR-10 with standard data augmentation (random cropping, flipping, etc.) and the student network is trained with them. The visual encoder $E_T$ is pretrained on the DINO method and kept frozen throughout training, while the focus is on learning the symbolic representations.

**Loss Function** The loss function is designed to guide the student model in learning from both continuous and symbolic representations. Building on the DINO loss, we introduce a granularity loss that accounts for varying levels of detail in the symbolic representations. This term adapts the local-to-global strategy from the DINO framework, encouraging the student to align its representations with those of the teacher across different levels of abstraction. The granularity factor allows the student to focus on both high-level and more detailed features of the input.

The overall loss function $\mathcal{L}_{SSL}$ is defined as:

$$\mathcal{L}_{SSL} = \sum_{i=1}^{V} \sum_{j=1}^{D} \lambda^j \times \mathcal{H}(\mathbf{p}_t^{(i)}, \mathbf{p}_s^{(i,j)})$$

where $\mathcal{H}(\mathbf{p}_t^{(i)}, \mathbf{p}_s^{(i,j)})$ is the cross-entropy between the teacher's visual representations $\mathbf{p}_t$ and the student's symbolic representations at different granularities $\mathbf{p}_s$, with $\lambda^j$ acting as a scaling factor for each level of detail.

**Optimization** We use the AdamW optimizer with gradient clipping (factor of 2) and mixed-precision training. The learning rate is scheduled similarly to DINO, and we train for 140 epochs with a batch size of 64 on a single GPU. A **k-NN probing** task is performed periodically to monitor the quality of the learned representations.

## 3.4 EXPLORATION STRATEGIES

We apply various exploration strategies to further enhance the diversity and richness of the symbolic sequences generated by the student network. These strategies are specifically used in the context of the Gumbel Softmax variation of the discretization process, where we investigate the effects of different loss terms to encourage exploration and variability in the symbolic sequences.

- **Base Strategy**: In the base strategy, we only modify the teacher temperature ($T$) and apply a scheduler to gradually decrease the Gumbel Softmax temperature ($\tau$), allowing the predictions to better approximate a maximum as training progresses. No additional loss terms are introduced in this baseline approach.

- **Entropy Encouragement Strategy**: Inspired by entropy-based exploration in Reinforcement Learning (e.g., SAC), we introduce a loss term that penalizes low entropy in the Gumbel Softmax predictions, encouraging the model to maintain higher entropy during training. This term is designed to foster more diverse symbolic sequences:

$$\mathcal{L}_{\text{entropy}} = -\alpha H(p),$$

where $H(p)$ is the entropy of the predicted sequence distribution, and $\alpha$ is a scaling factor.

- **Information Maximization Strategy**: We introduce another exploration strategy where a penalty is applied to sequences with low information content, measured using information theory. This encourages the model to produce sequences with high variability, avoiding the repetition of the same symbols. The relative or sampled information in the generated sequences is computed during training, and a penalty is applied based on the rate of symbol repetition:

$$\mathcal{L}_{\text{info}} = -\beta I(\mathbf{s}),$$

where $I(\mathbf{s})$ measures the information content of the symbolic sequence.

## 4 RESULTS

The evaluation of our symbolic representations centres on two key aspects: their effectiveness in downstream tasks and their interpretability.

**Probing Task** To assess the quality of the student network's representations during and after training, we first use a k-NN probing method, these evaluations are conducted on a classification task, using the test set labels solely as a metric for performance, varying systematically the number of neighbors (e.g., 10, 20, 100, 200). This provides an initial measure of how the representations cluster around meaningful patterns. After training, we employ a linear probing task to further assess the ability of these representations to encode useful and interpretable information. By training linear classifiers on the learned symbolic representations, we quantify the alignment of these features with human-understandable concepts, offering insights into their utility for downstream tasks.

Table 1: KNN and Linear Probing Results

| Model | KNN | | Linear | |
|---|---|---|---|---|
| | Top1 | Top5 | Top1 | Top5 |
| DINO | **91.08** | **99.35** | - | - |
| VQVAE 512 | 21.71 | 65.14 | - | - |
| VQVAE 256 | 26.37 | 72.55 | - | - |
| VQVAE 128 | 31.67 | 76.97 | - | - |
| Our(VQ 512) | 34.2825 | 88.365 | 0.3365 | 0.9030 |
| Our(VQ 256) | 20.7375 | 78.810 | 0.2082 | 0.7992 |
| Our(VQ 128) | 25.3150 | 80.0675 | 0.2202 | 0.8299 |
| Our(SS 512) | 31.6525 | 88.640 | 0.3335 | 0.9108 |
| Our(SS 256) | 10.1450 | 50.2425 | 0.0995 | 0.4976 |
| Our(SS 128) | 30.7925 | 86.6575 | 0.2868 | 0.8982 |
| Our(GS 512) | 49.1250 | 93.810 | 0.5248 | **0.9686** |
| Our(GS 256) | **50.0150** | **94.185** | **0.5301** | 0.9674 |
| Our(GS 128) | 47.3075 | 91.105 | 0.4983 | 0.9456 |

**Subsequence Analysis**    We evaluate the quality of symbolic representations by analyzing sub-sequences of varying lengths (e.g., 1-symbol, 2-symbol, etc.) and comparing three exploration strategies: Information, Entropy, and Base. As shown in Table 2, the Top1 and Top5 accuracies improve with increasing subsequence length, reflecting the compositional nature of the representations. For 1-symbol subsequences, the Information Strategy achieves 28.50% Top1 accuracy, while the Entropy and Base strategies score 27.13% and 27.52%, respectively. As subsequence length increases, all strategies improve significantly, with the 8-symbol sequences achieving Top1 accuracies of 41.95%, 42.43%, and 43.90%, respectively. This suggests that longer subsequences capture more meaningful information, and highlights the growing symbolic richness as sequence length increases.

Table 2: Subsequence Analysis on the different exploration strategies

| Subsequence Length | Information Strategy | | Entropy Strategy | | Base Strategy | |
|---|---|---|---|---|---|---|
| | Top1 | Top5 | Top1 | Top5 | Top1 | Top5 |
| 1 symbol(s) | 28.50 | 78.96 | 27.13 | 77.62 | 27.52 | 76.88 |
| 2 symbol(s) | 34.48 | 85.49 | 34.10 | 84.76 | 33.90 | 82.39 |
| 4 symbol(s) | 38.67 | 88.81 | 38.46 | 89.70 | 39.55 | 86.77 |
| 8 symbol(s) | 41.95 | 91.20 | 42.43 | 91.87 | 43.90 | 89.85 |

**Symbolic Interpretability**    Our approach provides interpretability by mapping discretized symbolic sequences to distinct visual features or concepts, which can be traced back to specific regions in the input image. To explore this interpretability, we generate attention maps that highlight these regions, allowing us to observe how the model encodes visual information. Through a qualitative analysis of symbolic tokens within specific classes, we observe consistent patterns in classes with lower visual variability, such as birds, where certain symbols, like the one shown in Fig. 5), often correspond to specific object parts. However, in more diverse classes like ships (not shown), the patterns are less distinct, with shared symbols frequently associated with background elements. The best results were achieved using a temperature-softmax discretization process, likely due to reduced noise during training. These observations provide insights into how the model organizes symbolic representations across different levels of intra-class variability.

## 5    DISCUSSION AND CONCLUSIONS

In this work, we explored a novel SSL approach to generate symbolic representations from visual data. Our experiments demonstrated the potential of combining discretization strategies with self-supervised learning to produce symbolic sequences that abstract visual information meaningfully. Despite several challenges, we observed promising results in both the interpretability and performance of our symbolic representations.

### 5.1    KEY INSIGHTS

**Symbolic Representation Effectiveness**    Our method successfully generated symbolic sequences that aligned with visual representations from a pretrained Vision Transformer. As seen in the probing tasks, the student model's symbolic representations performed well, especially in the Gumbel-Softmax discretization strategy. This indicates that symbolic abstraction can retain relevant information and facilitate downstream tasks like classification. However, we also observed that accuracy improves with sequence length, suggesting that compositionality is critical for capturing more detailed aspects of visual data.

**Interpretability of Symbolic Representations**    A core strength of our approach lies in the interpretability of the symbolic sequences. As shown in the qualitative analysis (Fig. 5), symbolic tokens consistently corresponded to specific visual concepts, particularly in low-variability classes like birds. This transparency is a significant step toward more explainable AI systems, as it allows us to trace how symbolic representations map to visual input. However, more diverse classes showed less consistency, underscoring the need for future work to handle high intra-class variability better.

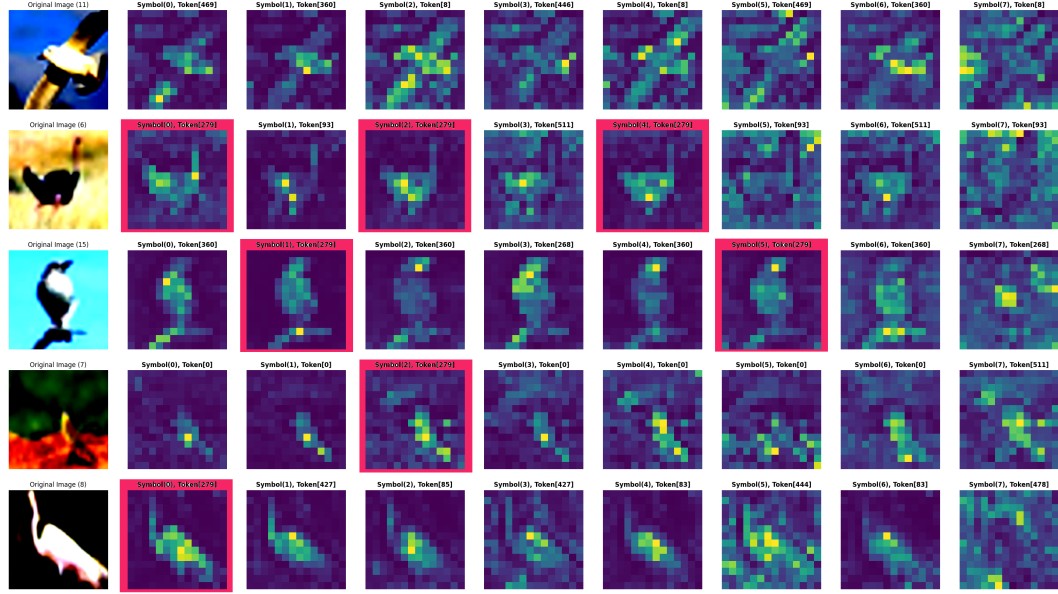

Figure 5: Qualitative analysis of symbolic interpretability for the "bird" class, focusing on the appearance of symbol 279 across multiple samples. The figure shows input images, followed by attention maps that highlight regions corresponding to symbol 279. This symbol consistently appears in specific locations across different samples, often linked to parts of the bird, such as the body. Such patterns are common in classes with low visual variability, like birds, whereas classes with higher variability (e.g., ships, not shown) exhibit more localized and less consistent behavior. All sequences and attention maps are generated using a temperature-softmax discretization with a temperature of 0.12.

## 5.2 LIMITATIONS AND FUTURE WORK

Despite the positive outcomes, our method faces several limitations:

**Training Constraints** Due to computational limits, we trained models for under 200 epochs, which may have constrained their performance. Additionally, training was limited to the CIFAR-10 dataset, which may restrict the generalizability of our findings. Future work should aim to train on larger datasets, such as CIFAR-100 or more complex synthetic datasets, to explore the scalability of our method.

**Challenges in Discretization and Exploration** We found that discretization strategies plateaued in performance around 50% accuracy, but this improved to 60% by using a combined strategy during training. Nonetheless, the Gumbel-Softmax approach introduced noise, limiting both interpretability and performance. Future work could focus on refining these discretization techniques and understanding how symbolic diversity correlates with model accuracy. Additionally, exploration strategies, while promising for increasing sequence variability, did not significantly boost performance, indicating that further research is needed to optimize this aspect.

## 5.3 CONCLUSIONS

This study highlights the viability of symbolic representations for visual data, offering a pathway to more interpretable models that maintain strong performance on downstream tasks. While there is room for improvement in terms of generalization and efficiency, the success of our approach in extracting meaningful symbolic information provides a foundation for future research into symbolic reasoning and representation in AI systems.

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
