# OpenReview forum: "Extracting Symbolic Sequences from Visual Representations via Self-Supervised Learning"
_ICLR.cc/2025/Conference — Submitted to ICLR 2025_

### Official Review · Reviewer_xeVQ · 2024-10-29

**Soundness:** 2
**Presentation:** 3
**Contribution:** 2
**Rating:** 3
**Confidence:** 4

**Summary:**

The paper proposes to abstract complex visual information into discrete, structured symbolic sequences using self-supervised learning (SSL). It extends the DINO framework to handle both visual and symbolic information. The method improves interpretability by linking the generated symbols to visual regions through attention maps.

**Strengths:**

1.	The paper focuses on an important and longstanding problem: extract symbolic information from complex visual input.
2.	The paper extend the DINO framework to handle both visual and symbolic information.

**Weaknesses:**

My main concerns are: lack of comparison with existing methods, and insufficient study about symbolic representations. The paper leverage several existing techniques (e.g., SSL, Vector Quantization), but it is confusing what is the new insight or discovery.

1.	Lack of comparison with existing methods. Tab.1 and Tab.2 are ablation studies about some strategy choices. However, how does the proposed model perform on image/video classification benchmarks? How is it compared with original DINO, or other SSL model without symbolic representation?
2.	Few explorations of the advantage/necessity of symbol representations. The main claimed advantage is interpretability, i.e., several discrete feature maps in Fig.5. However,
    - How can we use this interpretability in practice? Or, what are the conditions DINO feature fails to interpret? Since DINO feature has shown some emerging detection/segmentation ability.
    - How can the learned “symbols” benefit high-level scene understanding and abstract reasoning? The output feature maps are still complex and need human to summarize their meanings.

**Questions:**

My questions are listed in weakness section. My main concerns are performance comparison with existing SSL methods, and superiority of the proposed method.

---

> ### Author Response · Authors · 2024-11-29
>
> Thank you for your insightful feedback and the critical points you've raised regarding the comparison with existing methods and the exploration of symbolic representations. I have now included comparisons between our approach, VQVAE, and standard DINO representations.
>
> Regarding the value of symbolic representations, our primary focus has been on improving generalization rather than immediate applications. While DINO's perceptual features provide a strong foundation, we aim to transform them into compact, hierarchical symbolic representations that capture core concepts in a scene. These symbolic representations are designed to support abstract reasoning by offering a conceptual understanding rather than relying on perceptual details. Although there is still work to be done in aligning symbols with image segments, we anticipate that as the method evolves, the correspondence will improve, and these symbolic representations will become more interpretable and aligned with higher-level concepts.
>
> Once again, thank you for your thoughtful feedback. I'm happy to address any additional questions or concerns you may have.

---

> > ### Comment · Reviewer_xeVQ · 2024-12-02
> >
> > Thank you for updates and detailed feedback.
> >
> > "I have now included comparisons between our approach, VQVAE, and standard DINO representations.": Is it Tab.1？Why linear probing is missing for VQVAE/DINO, and why in KNN probing DINO performs better (since the metric seems to be classification accuracy)?
> >
> > My concerns about weakness2 still remains. More pratical, expressive results and discovery should be shown.
> >
> > Thus, I keep my rating.

---

### Official Review · Reviewer_Q7WH · 2024-10-30

**Soundness:** 1
**Presentation:** 1
**Contribution:** 1
**Rating:** 3
**Confidence:** 3

**Summary:**

The paper proposes a self-supervised learning method to learn visual symbolic representations. They use a teacher-student framework where the student predicts a discretised symbolic representation from source visual representations coming from a fixed teacher. The authors use attention masks used for predicted visual symbols to visualise what part of the image they activate on.

**Strengths:**

1. The motivation behind the approach and value of building systems that can extract symbolic representations from images is well presented.
2. The overall paper organisation is good (Although major issues in writing discussed below)

**Weaknesses:**

1. The experiments are very weak.
	* (a) The authors cite computational limitations but this work needs multiple datasets and even larger scale datasets for a decent validation of the approach.
	* (b) I am disappointed the authors do not experiment on any downstream tasks to illustrate the use of extracting symbolic representations. Some reasoning task (eg. VQA) or by-design interpretability via logical rules could be options. They will really drive home (1) the usefulness of visual symbolic representations, (2) that your method provides good quality of representations.
	* (c) There are no external baselines anywhere in the paper. I also wonder how the method would compare to a baseline if one simply learns a codebook by clustering DINO CLS token representations.
	* (d) No type of interpretability evaluation of attention masks. Some kind of segmentation or faithfulness evaluation should be possible. The authors should can check out the Quantus package for such metrics.

2. Throughout the system design and experiments, things are very poorly described, and are often unclear.
	* (a) The authors don't describe the system notations well. There are no notations for any input/output domains anywhere for any representation. At the bare minimum I'd expect you described dimensionality of your symbolic sequence/token (line 195, 201) but even that is not done. What is this teacher's joint distribution space "p"? Is it distribution of predicted class probabilities? Line 216 figure heading "KL Divergency" -> "KL Divergence".
	* (b) Table captions are barely descriptive. They do not describe what is inside the table. Even the main text is only marginally better at describing what is inside the tables. For instance in table 1, I do not understand what is the top-1, top-5 supposed to be for. Is it classification accuracy, which is what I am guessing right now? Is it accuracy for some underlying ground-truth set of human understandable concepts as lines 338-343 seem to indicate? The model abbreviations I assume are the different discretization strategies (should be clearly specified), but what is the number afterwards? Based on the paper I think it is the codebook size but nowhere in the whole section is this mentioned.

In my opinion, the paper requires major updates in experiments and writing.

**Questions:**

My questions asking for clarifications about writing are in Weaknesses 2(a), 2(b).

---

> ### Author Response · Authors · 2024-11-29
>
> Thank you for your detailed feedback and for highlighting key areas where improvements can be made. I understand your concerns regarding experimental validation and the need for more comprehensive evaluations, including the use of external baselines and downstream tasks. While we are still refining the method, we acknowledge that additional validation, particularly with larger datasets, will be essential to fully assess its performance. Our current focus is on enhancing the core aspects of the method, specifically the representation of information in sequences and their separability, which we believe are key to the model's future success.
>
> To address the concern about external baselines, I have included VQVAE as a reference, using similar configurations, such as vocabulary size. Regarding your suggestion about the CLS evaluation approach(if understood well it matches the standard DINO eval), we have replicated the KNN probing evaluation from the original DINO work and added the results to the table. We are working towards matching DINO's visual representations, requiring further refinement of the discretization process to achieve high-quality symbolic representations.
>
>
> In response to your feedback on downstream tasks, we are exploring Neural Machine Translation (NMT) tests to better understand the symbolic sequences, although the results are still partial and not yet ready for full presentation. We also appreciate your suggestion to explore the Quantus package for interpretability evaluation and will incorporate it into our validation pipeline for future work. Additionally, I apologize for the oversight regarding system notations and table descriptions; the necessary corrections have been made, though I mistakenly selected the wrong version of the file during the update.
>
> Thank you once again for your thoughtful and constructive feedback. I would be happy to address any further questions or clarifications you may have.

---

> > ### Comment · Reviewer_Q7WH · 2024-12-01
> >
> > Thank you for the updates. I sincerely wish you the best for revising the work. I also find Reviewer pw7M's suggestions as interesting directions to consider.

---

### Official Review · Reviewer_VG9J · 2024-11-02

**Soundness:** 2
**Presentation:** 2
**Contribution:** 2
**Rating:** 5
**Confidence:** 3

**Summary:**

This paper proposed a novel self-supervised learning approach to extracting discrete and structured symbolic sequences from visual data. The core of this approach is a teacher-student framework, where the teacher provides pretrained visual features, and the student learns to convert visual features into symbolic sequences. The proposed model is trained on a variant of DINO loss, and the authors apply various strategies to enhance the generated symbolic sequences. The authors conduct experiments to visualize the region of symbolics, suggesting that the generated symbolic sequences capture a meaningful level of abstraction.

**Strengths:**

The main advantage of the proposed method is interpretability. The decoder transformer allows the attention maps to be linked to specific symbols, which indicates the regions these representations attend. This helps create interpretable symbolic representations, which can be applied to high-level scene understanding tasks.

**Weaknesses:**

1. This paper lacks details on model implementation and experimental setup, such as the structure of the encoder and decoder, and how the KNN and Linear Probing experiments were conducted.
2. This paper appears to lack important baseline models. All experiments only use different variants of the proposed method, without including other relevant models.
3. In Fig. 2, the overview of the proposed approach is not clear. For instance, the figure does not illustrate which modules belong to the teacher models and which are the student models, and the projector is not shown in the teacher models.

**Questions:**

Could the authors provide test results of the proposed symbolic representations on high-level downstream tasks? Empirically, interpretable visual symbols could help address certain visual understanding tasks.

---

> ### Author Response · Authors · 2024-11-29
>
> Thank you for your valuable feedback and insights. In response to your comments, I have included VQVAE representations as a reference to better contextualize the proposed method and have improved the probing section for clarity.
>
> Additionally, we are currently working on evaluating our symbolic sequences using Neural Machine Translation (NMT) tests to further understand their potential in high-level tasks. However, these results are still partial and not yet finalized. I also apologize for not being able to improve Figure 2 due to time constraints; I understand its clarity could be better. Thank you again for your thoughtful feedback, and I would be happy to address any further questions or clarifications.

---

> ### Comment · Reviewer_VG9J · 2024-12-03
>
> Thank you for the response. Based on the current version of manuscript, I would like to maintain my score at 5.

---

### Official Review · Reviewer_pw7M · 2024-11-04

**Soundness:** 2
**Presentation:** 2
**Contribution:** 2
**Rating:** 5
**Confidence:** 4

**Summary:**

This paper proposes a method for generating structured symbols from visual data based on abstract language.

**Strengths:**

The motivation of this article seems reasonable; however, the authors need to clarify this further in the discussion section, as there are many issues related to the core contributions.

**Weaknesses:**

- The writing of this article is not clear, and there are significant problems with the experimental comparisons, lacking implementation and experimental details.
- The Teacher network is initialized with a ViT, and the attention structure of such self-supervised learning models fully has the capability to capture objects, as evidenced by the visualizations in many papers. Therefore, the images in Fig. 5 do not seem convincing.
- Additionally, most self-supervised visual models have the ability to discover representations. Similar to Fig. 5, I cannot understand what is special about the representation in this paper.

**Questions:**

What exactly is the symbolic sequence in this paper? Is it just a kind of VQVAE?

---

> ### Author Response · Authors · 2024-11-29
>
> Thank you for your thoughtful feedback and for raising important questions regarding our approach. While our method shares some conceptual similarities with VQVAEs, such as unsupervised learning and the use of quantized representations, it diverges significantly in its purpose and structure. Specifically: (1) our approach is designed to learn hierarchical representations, where individual symbols capture high-level, abstract descriptions of objects, and complete sequences provide increasingly granular and detailed depictions, and (2) unlike VQVAEs, which focus primarily on latent representations without the intention of generating interpretable abstractions, our method explicitly aims to produce compact symbolic representations. These representations resemble image-like descriptions, characterized by low vocabulary sizes and short sequence lengths, aligning with image-captioning architectures.
>
> Although our symbolic sequences are not yet fully formalized and lack grammar or syntax, they show potential for compositionality and offer interpretable mappings through cross-attention, aligning symbols with visual regions. This aspect could provide a pathway to bridging visual data and abstract symbolic reasoning in future iterations. Thank you again for your valuable feedback, and I would be happy to address any further questions or clarifications.

---

> > ### Comment · Reviewer_pw7M · 2024-11-30
> >
> > Thank you for the author's response. I have carefully read the comments from other reviewers as well as each of the author's replies.
> >
> > I understand that what the author is trying to express is a self-supervised method, which initially compresses images into symbol sequences similar to VQVAE. However, the difference is that it selects some key symbols as representations to achieve better compression.
> >
> > I believe the experiments in the paper have taken a wrong direction. Here are some experiments that are relatively low-cost and more suitable for proving the author's contribution:
> > (1) To verify the better compression of symbols, the author should conduct image reconstruction experiments and verify indicators such as SNR.
> > (2) To validate meaningful symbols, the author should analyze the symbol dictionary, looking for relationships such as clustering and consistency with words, rather than performing attention visualization. This is because the attention of any self-supervised vision model can now perform object recognition, which does not provide meaningful validation for models initialized with ViT.
> > (3) The author should not be limited to conventional downstream task comparisons, because fewer and more concise representations will almost always lose out to complete representations. This is evident in most representation learning. The author should find the most suitable scenarios for this approach, such as low computational cost or trustworthy AI.
> >
> > If my understanding is correct, I believe the motivation is commendable, but the current writing and experiments in this paper are too weak. If this were a journal submission, I would encourage the author to resubmit. I hope the author can improve this paper in the next version.

---

### Official Review · Reviewer_e42M · 2024-11-04

**Soundness:** 3
**Presentation:** 2
**Contribution:** 3
**Rating:** 5
**Confidence:** 3

**Summary:**

The authors propose extending the DINO framework to handle both visual and symbolic information in order to generate symbolic representations from visual data.  Their approach learns structured symbolic sequences that capture the underlying compositional nature of visual data and is aimed at improving interpretability and visual reasoning tasks.

**Strengths:**

The paper is well motivated and the problem is important and strongly presented.  The writing is generally clear and easy to follow and the methods are evaluated clearly and against SOTA approaches.

**Weaknesses:**

Figure 2 should more explicitly point out which parts belong to the teacher and student models.  It would also help to have the figure match the description in 3.1 with explicit decoder, encoder and projector components labeled.

I am a little confused by the terminology and component descriptions in 3.1.  My understanding of the approach is that the student is trained as an auto-encoder that learns to encode continuous representations to discretized tokens.  The tokens are then converted to token representations that can be aligned with the original non-tokenized representation through a joint projection step.  Correct me if I am wrong.  I feel that the use of decoder and encoder in 3.1 is a little confusing as they are used differently to how I described above.  Perhaps cleaning it up as describing a decoder-encoder model that is aiming to learn a discretized decoding of the continuous latent space would be more clear.

On line 198, How does D_s discretize the latent?  This isn't clear in the next section either.

On line 235, how are the tokens defined?  This hasn't been covered yet.  This section is about selecting tokens in the discretization process but does not give an overview of what those tokens are or how they are computed.  If this is in prior work, it would be helpful to give a brief technical overview of the process used, with citations for the source of the approach, so the paper can be more self contained and reproducible.

Tables and results should include some measure of uncertainty to properly judge the significance of the results.

**Questions:**

See weaknesses.

---

> ### Author Response · Authors · 2024-11-29
>
> Thank you for your detailed feedback and for highlighting key areas for improvement. I have carefully revised the text to address concerns related to clarity and terminology. The methods section has been expanded to better explain the roles of the encoder, decoder, and projector components. Additionally, I have refined how the model discretizes latent representations and computes tokens, with a more detailed overview.
>
> Although I have made these textual adjustments, I regret that, due to time constraints, I could not update Figure 2.
>
> Thank you once again for your insightful feedback and perspective. I would be happy to address any further questions or concerns you may have.

---

> > ### Comment · Reviewer_e42M · 2024-12-03
> >
> > Thank you for the response and the description of the improvements to the paper.

---

### Meta-Review · Area_Chair_2ALz · 2024-12-20

**Metareview:**

This work proposes a new self-supervised learning approach to extend the DINO framework generate "symbolic sequences". The model is trained using a teacher-student framework from a pre-trained visual encoder, into a student network with a discretized representation space, trained using a variation of the loss from DINO. The model is evaluated using KNN and linear probing of the learned representation on a classification task, as well as qualitative results for interpretability. Reviewers raised key concerns on the experiments, missing details in the methods, and lack of baseline approaches. These concerns remain after rebuttal and there is unanimous decision to reject the paper from reviewers. The AC agrees and encourage the authors to take into account the valuable feedback from reviewers when making revisions to their work.

**Additional Comments On Reviewer Discussion:**

Reviewers had concerns on the method description, experimental design, and lack of baseline approaches. After rebuttal, these concerns remain, and there is agreement to reject this work. Incorporating suggestions from reviewers with a more complete experimental section would be crucial for future versions of this work.

---

### Decision · Program_Chairs · 2025-01-22

Reject